# Impact of the COVID-19 pandemic on women in the workplace in the Middle East and North Africa: A scoping review protocol

**Malak Ghezzawi**[1], **Sasha Fahme**[1], **Reem Al Sabbagh**[1], **Salpy Naalbandian**[2], **Jocelyn DeJong**[1]*, **WOMENA Study Group**¶

**1** Faculty of Health Sciences, American University of Beirut, Beirut, Lebanon, **2** University Libraries (Science & Agriculture Library), American University of Beirut, Beirut, Lebanon

¶ Membership of the WOMENA Study Group is provided in the Acknowledgments.

* jd16@aub.edu.lb

## Abstract

### Introduction

The COVID-19 pandemic has disrupted the livelihoods of working men and women world-wide. The pandemic exacerbated already existing inequities, especially in sectors where women predominate, such as the healthcare, education, and hospitality sectors. Women in the Middle East and North Africa (MENA) region, a world outlier for low female labor force participation despite high female education levels, may have been disproportionately impacted by the pandemic. Understanding the impact of COVID-19 on women's livelihoods and health is critical to support and retain women in the labor force during external health shocks. However, until now there has been relatively little research on this topic in MENA. Therefore, this scoping review aims to determine the impact of the COVID-19 pandemic on the health and wellbeing of women workers in low-and-middle-income countries in the MENA region.

### Methods

The outcome of interest is COVID-19 related occupational health (COVID-19 infection related to workplace exposures and pandemic-related occupational stressors) and the impact on mental and physical health among women workers in LMIC in MENA. Academic databases, including APA PsycINFO, Arab World Research Source: Al-Masdar, Global Health, MEDLINE, Scopus and Web of Science Core Collection will be searched. The study selection process will involve two independent reviewers and data extraction will involve summarizing key information from the included studies using a predefined charting table. The evidence will be analyzed descriptively, providing a comprehensive overview of the identified themes and patterns.

### Discussion

It is anticipated that this review will facilitate a deeper understanding of the impact of the COVID-19 pandemic on working women in the MENA region. The findings may inform data-

**Data Availability Statement:** No datasets were generated or analysed during the current study. All

relevant data from this study will be made available upon study completion.

**Funding:** This study is supported by a two-year grant from the International Development Research Centre (IDRC) in Canada under the reference number 110025. The funders had no role in study design, data collection and analysis, decision to publish, or preparation of the manuscript.

**Competing interests:** The authors declare that they have no competing interests.

**Abbreviations:** MENA, Middle East and North Africa; LMICs, low- and middle- income countries; PRISMA-ScR, PRISMA guidelines for Scoping Reviews; PRISMA-P, PRISMA guidelines for protocols; ECD, Early childhood development.

driven policies and targeted interventions that not only attract and retain women in the workforce but also enhance their health and well-being.

## Introduction

The Middle East and North Africa (MENA) region is a global outlier for low female labor force participation despite high levels of women education, a phenomenon known as the "MENA paradox" [1]. In 2022, women accounted for 19.8% of the labor force in the MENA region, compared with 39.5% globally [2]. This paradox has been explored largely by focusing on restrictive gender norms, the undervaluation of women's work particularly in relation to women's reproductive roles, and longstanding lack of investment in early childhood development services, all of which impede the economic participation and advancement of women, who are typically the primary caregivers [1, 3, 4]. Notably, women constitute a sizable share of the healthcare and education sectors in the MENA region, with 38.3% of employment being female in the health and social sectors [5]. This is significantly higher compared to the 15.6% share of female employment across all sectors [5]. This substantial representation highlights the critical impact of the COVID-19 pandemic on these sectors, which have been disproportionately affected [5, 6].

The MENA region is additionally characterized by conservative social norms and patriarchal hegemonic structures which shape women's societal roles [1]. Protracted conflicts impacting multiple low-and-middle-income countries (LMICs) in the region have resulted in the largest numbers of forcibly displaced individuals globally, introducing added layers of complexity to the socioeconomic landscape [7–9]. The backdrop of violence and instability impacts women's economic prospects, rights, and ability to access justice systems [7–9].

Women workers globally face gendered pay gaps and challenges reconciling caregiving responsibilities with professional obligations [10–12]. These limitations affect women's employment experiences, with consequences on their overall health and well-being [3, 13–15]. The impact of workplace stressors, hazards, and exposures on women's health are underreported and poorly characterized [16–18]. Examining gendered occupational health outcomes is critical to identify potential unmet health needs of female workers, improve their health, and support their retention in the workplace.

Regional evidence from MENA suggests additional barriers to employment beyond gender gaps in payment. Research from Jordan, for example, indicates that higher wages may induce participation, but that mixed-sex workplaces remain an important deterrent [19], while employer surveys from Egypt report explicit employer discrimination [20].

The COVID-19 pandemic added to these pre-existing disparities [21]. A large proportion of healthcare workers, the majority being women, faced a heightened risk of COVID-19 exposure and mental health challenges due to their patient-care duties in stressful working conditions [22, 23]. In Wuhan, China, for example, women healthcare professionals were more likely to develop depression during the pandemic than their male counterparts [24]. Similarly, in Turkey and India female healthcare personnel experienced higher levels of COVID-19-related stress than men [25, 26]. Women working in sectors like education also encountered distinct challenges [27, 28]. With school closures and the shift to remote learning associated with 'lockdowns', female educators had to swiftly adapt to address the "double burden" of caregiving and teaching, with detrimental mental health effects [29–32].

In this review, the focus is on COVID-19 related occupational health outcomes and its impact on the mental and physical health among women workers, an overrepresented heterogenous population in the health, education, and early childhood development (ECD) sectors, across LMICs in the MENA region. Women entering the workforce in LMICs in MENA are hindered due to the stereotypical and cultural roles ascribed to them and fewer socioeconomic protections, and the latter are also impacted by the ongoing political conflicts in the region [1, 7–9]. This review is being conducted as part of a larger project on women's health and economic participation in MENA that includes both primary and secondary data collection analysis.

## Objectives

The aim of this scoping review protocol is to explore the literature on how the COVID-19 pandemic affected working women's health and wellbeing across LMICs in the MENA region from January 2020 to August 2024. The review will address the following specific objectives:

1. Primary Objective: To map the range of mental and physical health outcomes experienced by women workers during the COVID-19 pandemic, providing a comprehensive overview of its impact on their health and overall well-being.

2. Secondary Objectives:

   - To evaluate the nature and quality of the evidence concerning the impact of COVID-19 on women workers, including study designs (e.g., cross-sectional, longitudinal, trials).

   - To evaluate the representation and experiences of diverse populations of women workers, including migrant workers and refugees.

   - To inform the development of research methodologies for a broader project on women's health and economic participation in MENA by elucidating the distinct health challenges faced by women workers during the pandemic.

   - To identify gaps in the current evidence base and highlight areas requiring further research.

## Methods

A scoping review will be undertaken to compile hits from numerous sources, from several peer-reviewed databases. This scoping review protocol was developed in consultation with an information specialist with expertise in library science. The review will adhere to the Preferred Reporting Items for Systematic Reviews and Meta-Analyses extension for Scoping Reviews (PRISMA-ScR) [33], PRISMA guidelines for protocols (PRISMA-P) [34], and the updated methodological guidance for the conduct of scoping reviews [35]. The review will consider all peer-reviewed publications which include primary data and assess the impact of COVID-19 on health outcomes among women workers and their wellbeing in English, Arabic and French. All study types, including case-control, retrospective and prospective cohort, cross-sectional, qualitative, mixed-methods, and interventional studies will be included.

## Eligibility criteria

### Inclusion criteria

The scoping review will focus on literature in peer-reviewed journals. The review will consider studies conducted in all 14 MENA countries which meet World Bank criteria as low- or

middle-income, being Algeria, Djibouti, Egypt, Iraq, Jordan, Lebanon, Libya, Morocco, Palestine, Somalia, Sudan, Syria, Tunisia, and Yemen. The review will encompass women workers across diverse sectors, including health, education, and ECD. Workers in both the public and private sectors will be included. Both pregnant and non-pregnant women workers will be considered, including formal and informal workers across different occupational strata to understand differences in types of exposures and of health outcomes. The outcome of interest being COVID-19 related occupational health is broadly defined to include COVID-19 infection related to workplace exposures, as well as pandemic-related occupational stressors. Studies conducted among populations within the countries of interest on the basis of ethnicity, religion, or country of origin will not be excluded, as the review aims to examine experiences of diverse populations of women workers, including migrant workers and refugees. Studies conducted in English, French, and Arabic will be included to enhance the diversity and inclusivity of the literature examined.

## Exclusion criteria

Studies which do not meet inclusion criteria, including those conducted within high-income MENA and non-MENA countries, will be excluded. Studies which do not report primary data, including systematic reviews, meta-analyses, book chapters, and commentaries will be excluded. Finally, as this review is restricted to the peer-reviewed literature, grey literature and conference abstracts will be excluded.

## Information sources and search strategy

The search will encompass all peer-reviewed articles published between January 2020 to August 2024. Six databases (MEDLINE (OVID), Arab World Research Source: Al-Masdar, APA PsycINFO (EBSCO), Global Health (CAB), Scopus, Web of Science Core Collection) will be used for this review. Refer to **S1 Appendix** for the MEDLINE search strategy. The search strategies will be developed in collaboration with a librarian to ensure thorough and effective retrieval of relevant literature.

## Data management

Studies retrieved from the search will be imported into Covidence software for de-duplication and screening.

## Selection process

Reviewers will independently assess each title and abstract in duplicates to determine its relevance to the review question and inclusion criteria. The review process will be visually presented in a flowchart, adhering to the PRISMA-ScR statement. The protocol will include separate appendices to provide details of included sources and a brief mention of the excluded sources. For excluded sources, the reasons for exclusion will be clearly stated, ensuring transparency and reproducibility in the review process.

Before commencing the actual source selection process across the entire team, a pilot testing phase will be conducted to ensure consistency and clarity in applying the eligibility criteria and then the team will meet to discuss any discrepancies or uncertainties that arise during the pilot testing. Based on the discussions and feedback, modifications to the eligibility criteria and definitions/elaboration document will be made to ensure clarity and consistency. The team will only begin the full source selection process when a consensus agreement of 95% or greater is achieved among the reviewers during the pilot testing. The selection process will

occur in two stages: title and abstract examination, followed by full-text examination. Reviewers will begin by screening the titles and abstracts of all identified records from the search in duplicates using the pre-defined inclusion criteria. After the initial title and abstract screening, reviewers will obtain the full-text articles of the potentially relevant records. The reviewers will then independently examine the full-text articles in duplicates to assess their eligibility for inclusion in the scoping review. In cases where there are disagreements between the reviewers regarding the inclusion of a particular study, the reviewers will attempt to reach a consensus through discussion and mutual agreement. If a consensus cannot be reached, a third reviewer will be involved to make the final decision on the inclusion or exclusion of the study.

## Data collection and extraction process

A draft charting table (**S2 Appendix**) will be developed at the protocol stage to record key information from each source. The charting table will include the following categories: identifying information (study citation), study design and methodology, aim of the study, participant characteristics, study outcomes/results, interventions of interest, quality assessment, conclusions, and implications. Data charting will be done by two members of the review team. Pilot testing of the charting table on a few selected sources will also take place at this stage. Before starting the full source selection process, the review team will ensure that 95% or greater agreement is achieved during the pilot testing phase to ensure that the charting process is aligned among the reviewers and that the table accurately captures all relevant information. The charting table will be updated continually as additional data are charted, ensuring comprehensive coverage of relevant information from each source. Reviewers will be open to extracting any relevant data that aligns with the scoping review questions, even if not initially included in the charting table. The data will be presented in tables and charts representing the prevalence of specific health outcomes among different groups of women workers in each country, and others summarizing key findings related to each health outcome. In addition to diagrammatic or tabular representations, the results will also be presented in a descriptive format.

## Data analysis

Data analysis will involve descriptive mapping of the extracted results from the included sources. The focus will be on providing a summary of the key concepts, populations, characteristics, and outcomes related to women workers' health and wellbeing during the COVID-19 pandemic.

## Ethical considerations

As this review involves a synthesis and presentation of available resources, it does not require ethics approval. Results will be published in a peer-reviewed journal, developed into easily disseminated infographics and shared at international conferences.

## Status and timeline of the study

A search strategy was developed for MEDLINE in consultation with an information specialist. It is anticipated that implementing the search on MEDLINE along with other databases, followed by title and abstract- and full-text-screening, and data extraction will happen within three months. Data analysis and writing will be conducted concurrently.

## Discussion

There is currently little to no data available on the impact of the COVID-19 pandemic on women workers' health and wellbeing in the MENA region, particularly within LMICs. This gap is evident in the available scoping reviews, which primarily focus on global populations, often excluding the MENA region [36–40]. The exclusion of this region is particularly important given its unique socioeconomic and cultural challenges, including higher rates of informal employment, gendered labor market segregation, and limited access to healthcare and social protections for women [1, 3–5].

For instance, systematic reviews to date have predominantly concentrated on mental health outcomes [41–45], COVID-19 related mortality rates [46], and clinical outcomes such as morbidity and mortality among healthcare workers [47]. One notable scoping review from Canada examined stress, burnout, and depression among women healthcare workers during the COVID-19 pandemic, identifying stressors at the individual, organizational, and systemic levels [48]. This review highlighted the scarcity of effective interventions and revealed the heightened vulnerability of women healthcare workers to psychological stress and burnout, particularly among younger and mid-career professionals, those without social support, and those with medical comorbidities or increased alcohol consumption [48]. Although this review included Saudi Arabia and Turkey, its scope was not exclusive to the MENA region, and it did not capture the broader experiences of women in non-healthcare occupations and their overall well-being.

Existing studies, while valuable, do not capture the full spectrum of the pandemic's impact on women workers in MENA, particularly in terms of non-healthcare-related occupations, employment stability, and broader well-being. This scoping review aims to fill this significant gap by providing a comprehensive analysis of how the COVID-19 pandemic has affected working women in the MENA region. By focusing on the intersection of gender, occupational roles, and the pandemic, the review will offer nuanced insights that are currently lacking in the literature. The findings will provide valuable evidence to inform data-driven policies and interventions tailored to the needs of women workers in MENA, ultimately supporting efforts to attract and retain women in the workforce while enhancing their overall health and well-being. In turn, this can contribute to broader economic and social development goals within the region.

## Limitations

Relying solely on specific databases for the literature search may cause pertinent peer-reviewed publications that are not indexed in these databases to be overlooked. To mitigate this limitation, the included papers will be cross-referenced to identify additional relevant studies. Additionally, our analysis only considers peer-reviewed literature, leaving out potentially insightful information from sources of grey literature such reports, dissertations, and conference proceedings. This exclusion could result in the removal of essential research that are not well publicized or a bias towards published efforts.

## Dissemination plans

Results will be published in a peer-reviewed journal and shared at international conferences. During the process of this review, the team has ongoing engagement with local and regional public sector actors, thus the results of this review will be disseminated to them as well.

## Supporting information

**S1 Checklist. PRISMA-P (Preferred Reporting Items for Systematic review and Meta-Analysis Protocols) 2015 checklist: Recommended items to address in a systematic review**

protocol*.
(DOC)

**S1 Appendix. Search strategy.**
(DOCX)

**S2 Appendix. Data extraction instrument.**
(DOCX)

## Acknowledgments

We would like to thank the WOMENA Study Group for supporting the study work (Stephen McCall, Hala Ghattas, Rita itani, Myriam Dagher, Ali Abboud, Nisreen Salti, Ghada Saad).

## Author Contributions

**Conceptualization:** Sasha Fahme, Jocelyn DeJong.

**Funding acquisition:** Sasha Fahme, Jocelyn DeJong.

**Methodology:** Malak Ghezzawi, Sasha Fahme, Salpy Naalbandian, Jocelyn DeJong.

**Supervision:** Sasha Fahme, Jocelyn DeJong.

**Writing – original draft:** Malak Ghezzawi, Reem Al Sabbagh.

**Writing – review & editing:** Malak Ghezzawi, Sasha Fahme, Salpy Naalbandian, Jocelyn DeJong.

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
