## [Decision Letter · Decision Letter 0]

13 Aug 2024

PONE-D-24-14479Impact of the COVID-19 Pandemic on Women in the Workplace in the Middle East and North Africa: A Scoping Review ProtocolPLOS ONE

Dear Dr. DeJong,

Thank you for submitting your manuscript to PLOS ONE. After careful consideration, we feel that it has merit but does not fully meet PLOS ONE’s publication criteria as it currently stands. Therefore, we invite you to submit a revised version of the manuscript that addresses the points raised during the review process.

We look forward to receiving your revised manuscript.

Kind regards,

Olushayo Oluseun Olu

Academic Editor

PLOS ONE

“This study is supported by a two-year grant from the International Development Research Centre (IDRC) in Canada under the reference number 110025.”

Reviewers' comments:

Reviewer's Responses to Questions

**Comments to the Author**

1. Does the manuscript provide a valid rationale for the proposed study, with clearly identified and justified research questions?

Reviewer #1: Yes

Reviewer #2: No

Reviewer #3: Yes

2. Is the protocol technically sound and planned in a manner that will lead to a meaningful outcome and allow testing the stated hypotheses?

Reviewer #1: Yes

Reviewer #2: No

Reviewer #3: Yes

3. Is the methodology feasible and described in sufficient detail to allow the work to be replicable?

Reviewer #1: Yes

Reviewer #2: No

Reviewer #3: Yes

4. Have the authors described where all data underlying the findings will be made available when the study is complete?

Reviewer #1: No

Reviewer #2: No

Reviewer #3: Yes

5. Is the manuscript presented in an intelligible fashion and written in standard English?

Reviewer #1: Yes

Reviewer #2: Yes

Reviewer #3: Yes

6. Review Comments to the Author

You may also provide optional suggestions and comments to authors that they might find helpful in planning their study.

Reviewer #1: There is no explanation for data sets as this is a scoping review. My Comments have been uploaded in the attached document.

Reviewer #2: The manuscript does not effectively convey a clear protocol and and what its relevance and contribution to knowledge will be.

Reviewer #3: This manuscript titled “Impact of the COVID-19 Pandemic on Women in the Workplace in the Middle East and North Africa: A Scoping Review Protocol “ is well written and its methodology well detailed. It should prove useful for policy in the MENA region and add useful data and insight to research in a much needed area.

I however ,have listed few points that the writers need to resolve below.

1)Notably, women constitute an sizable share of the healthcare and education sectors in MENA, two sectors disproportionately impacted by the COVID-19 pandemic (5, 6).

What is this share? -elaborate with numbers and statistics for better clarity.

Grammar correction for the underlined highlighted words above- a sizeable share

2)The review will consider studies conducted in all MENA countries which meet World Bank criteria as low- or middle-income, including Algeria, Egypt, Iraq, Jordan, Lebanon, Libya, Morocco, Palestine, Sudan, Tunisia, and Yemen.

List out all the countries eligible for the study expressly= do not use including…

3)Ethical consideration. Ethical approval is not required for this review of the published, peer-reviewed literature.

Submission to ethic review is important and gives confidence that a local oversight role is available and applied . I would suggest submission for ethics consideration; an official documented waiver can then be granted . This waiver is quoted in lieu of ethical approval.

7. PLOS authors have the option to publish the peer review history of their article (what does this mean?). If published, this will include your full peer review and any attached files.

Reviewer #1: No

Reviewer #2: No

Reviewer #3: No

---

## [Author Response · Author response to Decision Letter 0]

23 Sep 2024

Journal: PLOS One

PONE-D-24-14479

 23-09-2024

Re: Manuscript entitled “Impact of the COVID-19 Pandemic on Women in the Workplace in the Middle East and North Africa: A Scoping Review Protocol

Dear Editor, 

On behalf of all co-authors, we would like to thank you and the reviewers for the time spent on reviewing our work, as well as the valuable comments that helped us improve the quality of our manuscript”. All comments have been addressed, as can be seen in the point-by-point response to the comments by reviewers below. Following each of the reviewers’ comments we have responded to each point that the reviewer makes that requires revision. 

All authors confirm that the manuscript is not under consideration in any other journal and that no conflict of interest exists. All authors have read and approved the submission of this manuscript.

Yours sincerely, 

Jocelyn DeJong, PhD

Associate Provost and Professor at the Faculty of Health Sciences, 

American University of Beirut, Beirut, Lebanon 

Email: Jd16@aub.edu.lb

Reviewer #1

We would like to thank the Reviewer for the time spent on our work and the valuable comments.

Summary Of review: 

The protocol is describing the study design for a study that the authors intend to undertake synthesizing the evidence on occupational health outcomes for women workers in the MENA region and their impact on the health (mental and physical) of the women. This is a very important research question that will enable the identification of regional priorities for occupational health for women in MENA, an important barrier to participation in the workforce. 

Background and context:

The general context of the study is well explained, outlining the global and regional evidence on women occupational health and the related challenges arising from COVID 19 at a global level. It also presents the general landscape of female participation in the workforce and the related health challenges. Given the fact that COVID 19 in general exacerbated many health-related aspects, the decision to conduct the scoping review is indeed valid. 

In terms of objectives of the study: 

The primary outcome of the study is the mental and physical health wellbeing of women workers. The objectives could be nuanced a bit given that it is informing the research methods of a larger project. It would be good to include an objective on the state of evidence itself. The types of studies that have been done and the quality therein. It should also identify the gaps in evidence if any. The authors state in the inclusion criteria that “Studies conducted among populations within the countries of interest on the basis of ethnicity, religion, or country of origin will not be excluded, as the review aims to examine experiences of diverse populations of women workers, including migrant workers and refugees. This kind of nuance should be brought formally upfront as part of the specific objectives and secondary outcomes of interest in the review. 

The exclusion criteria are well spelt out and comprehensive enough to establish the state of the evidence relevant to the primary objectives and secondary objectives including the impact on the diversity in the population. 

The time range on the study: Since some cases of post COVID syndrome/Long COVID have been known to appear months after the case, it might be prudent to have a longer period to capture any study that highlights cases of mental and physical health challenges that arose after the pandemic ended.

The methods adopted for study selection, data extraction and process are aligned with PRISMA guidelines. The use of COVIDENCE is appropriate for dealing with duplicates.

The Data analysis plan is also clear and concise. 

Overall Recommendation: Consider for publication with minor changes summarized below. I need not review.

• Elaborate the objectives of the study and nuance them to reflect the entire scope of what the study intends to achieve as highlighted above.

Thank you for your valuable feedback. We have revised the objectives of our scoping review to include a focus on the state of evidence, historical context, and identification of evidence gaps. The manuscript has been edited accordingly to reflect these updates.

• Reconsider the time limits of the study to reflect the possibility of post Covid syndrome and studies that have documented these outcomes.

Thank you for your valuable feedback. We appreciate your observation regarding the timeframe for including studies related to the pandemic. We had chosen this cut-off date because it was 3 months after the official end of the pandemic, as declared by WHO. In response to your comment, we agree that extending the search period would provide a more comprehensive inclusion of relevant studies, particularly those with longitudinal and comparative data that may have been published after the pandemic's official end. We will therefore revise our search strategy to include studies published up until August 2024. This extension will ensure that our review captures the most recent and relevant research related to the pandemic and its impacts. For practical reasons due to research funding, we are not able to extend the period beyond that. The manuscript has been edited accordingly.

Reviewer #2

We would like to thank the Reviewer for the time spent on our work and the valuable comments.

The manuscript does not effectively convey a clear protocol and what its relevance and contribution to knowledge will be.

Thank you for your feedback. We appreciate your concerns and would like to clarify the contributions of this protocol. This protocol is designed to systematically explore the impact of COVID-19 on the health and well-being of women workers in the MENA region. By adhering to established methodological guidelines such as PRISMA-ScR and PRISMA-P, the review ensures rigor and transparency throughout the process, which is crucial for generating reliable and actionable insights.

There is a notable lack of comprehensive data on women workers' health outcomes during the COVID-19 pandemic in the Middle East and North Africa (MENA), especially within low-and-middle-income countries (LMICs). The available scoping and systematic reviews either exclude the MENA region from global analyses or fail to provide an in-depth exploration of the diverse experiences of women workers across different occupational strata. This omission may be due, in part, to the fact that research from the MENA region is often published in non-English languages, making it less likely to be included in global reviews. Additionally, existing studies often lack a focus on the specific socioeconomic and cultural challenges that uniquely impact women workers in the region. This scoping review aims to fill these gaps by centering on the distinct contexts of women workers in the MENA region.

The findings from this review are expected to provide valuable evidence that can inform policies and interventions aimed at improving the health and well-being of women workers in the MENA region. By addressing the challenges these women face, the review will support efforts to attract and retain women in the workforce, ultimately improving the health and well-being of women in the region.

The manuscript has been revised accordingly to better reflect these points. 

Reviewer #3

We would like to thank the Reviewer for the time spent on our work and the valuable comments.

This manuscript titled “Impact of the COVID-19 Pandemic on Women in the Workplace in the Middle East and North Africa: A Scoping Review Protocol “ is well written and its methodology well detailed. It should prove useful for policy in the MENA region and add useful data and insight to research in a much-needed area. 

I, however, have listed few points that the writers need to resolve below.

1) Notably, women constitute an sizable share of the healthcare and education sectors in MENA, two sectors disproportionately impacted by the COVID-19 pandemic (5, 6). 

What is this share? -elaborate with numbers and statistics for better clarity.

Thank you for your feedback. We have added specific statistics on the share of women in the healthcare and education sectors in the MENA region to clarify their significant representation.

Grammar correction for the underlined highlighted words above- a sizeable share

Thank you for your comment. The grammar has been corrected for the underlined word.

2)The review will consider studies conducted in all MENA countries which meet World Bank criteria as low- or middle income, including Algeria, Egypt, Iraq, Jordan, Lebanon, Libya, Morocco, Palestine, Sudan, Tunisia, and Yemen.

List out all the countries eligible for the study expressly= do not use including…

Thank you for your comment. The protocol has been edited accordingly to explicitly list all eligible countries for the study. The review will consider studies conducted in the MENA countries that meet the World Bank criteria as low- or middle-income: Algeria, Egypt, Iraq, Jordan, Lebanon, Libya, Morocco, Palestine, Sudan, Tunisia, and Yemen. Additionally, we have now included Djibouti, Somalia, and Syria to ensure that all low- and middle-income countries in the region are covered.

3)Ethical consideration. Ethical approval is not required for this review of the published, peer-reviewed literature. 

Submission to ethic review is important and gives confidence that a local oversight role is available and applied. I would suggest submission for ethics consideration; an official documented waiver can then be granted. This waiver is quoted in lieu of ethical approval. 

Thank you for your comment regarding the ethical considerations for our scoping review protocol. As this review involves the synthesis and presentation of existing, publicly available resources from peer-reviewed literature, it does not require ethics approval. The nature of the study does not involve new data collection, interaction with human participants, or any procedures that typically necessitate ethical oversight. The Ethical Considerations section has been edited accordingly.

---

## [Editor Report · Decision Letter 1]

30 Sep 2024

Impact of the COVID-19 Pandemic on Women in the Workplace in the Middle East and North Africa: A Scoping Review Protocol

PONE-D-24-14479R1

Dear Dr. DeJong,

We’re pleased to inform you that your manuscript has been judged scientifically suitable for publication and will be formally accepted for publication once it meets all outstanding technical requirements.

Kind regards,

Olushayo Oluseun Olu

Academic Editor

PLOS ONE
---

## [Editor Report · Acceptance letter]

18 Oct 2024

PONE-D-24-14479R1 

PLOS ONE

Dear Dr. DeJong, 

I'm pleased to inform you that your manuscript has been deemed suitable for publication in PLOS ONE. Congratulations! Your manuscript is now being handed over to our production team.

Kind regards, 

on behalf of

Dr. Olushayo Oluseun Olu 

Academic Editor

PLOS ONE